

# The nematode homologue of Mediator complex subunit 28, F28F8.5, is a critical regulator of *C. elegans* development

Markéta Kostrouchová[1,2], David Kostrouch[1], Ahmed A. Chughtai[1], Filip Kaššák[1], Jan P. Novotný[1], Veronika Kostrouchová[1], Aleš Benda[3], Michael W. Krause[4], Vladimír Saudek[5], Marta Kostrouchová[1] and Zdeněk Kostrouch[1]

[1] Biocev, First Faculty of Medicine, Charles University, Prague, Czech Republic
[2] Department of Pathology, Third Faculty of Medicine, Charles University, Prague, Czech Republic
[3] Imaging Methods Core Facility, BIOCEV, Faculty of Science, Charles University, Prague, Czech Republic
[4] Laboratory of Molecular Biology, National Institute of Diabetes and Digestive and Kidney Diseases, National Institutes of Health, Bethesda, MD, USA
[5] Metabolic Research Laboratories, Wellcome Trust-Medical Research Council Institute of Metabolic Science, University of Cambridge, Cambridge, UK

Corresponding author
Zdeněk Kostrouch,
Zdenek.kostrouch@lf1.cuni.cz

## ABSTRACT

The evolutionarily conserved Mediator complex is a critical player in regulating transcription. Comprised of approximately two dozen proteins, the Mediator integrates diverse regulatory signals through direct protein-protein interactions that, in turn, modulate the influence of Mediator on RNA Polymerase II activity. One Mediator subunit, MED28, is known to interact with cytoplasmic structural proteins, providing a potential direct link between cytoplasmic dynamics and the control of gene transcription. Although identified in many animals and plants, MED28 is not present in yeast; no bona fide MED28 has been described previously in *Caenorhabditis elegans*. Here, we identify bioinformatically F28F8.5, an uncharacterized predicted protein, as the nematode homologue of MED28. As in other Metazoa, F28F8.5 has dual nuclear and cytoplasmic localization and plays critical roles in the regulation of development. *F28F8.5* is a vital gene and its null mutants have severely malformed gonads and do not reproduce. F28F8.5 interacts on the protein level with the Mediator subunits MDT-6 and MDT-30. Our results indicate that F28F8.5 is an orthologue of MED28 and suggest that the potential to link cytoplasmic and nuclear events is conserved between MED28 vertebrate and nematode orthologues.

## INTRODUCTION

The Mediator complex is a multiprotein assembly that is capable of integrating cellular signals with the regulation of transcription through direct interaction with RNA Polymerase II (Pol II). The Mediator complex is found in all eukaryotic organisms.

The core Mediator complex is comprised of 21 protein subunits in yeast and a similar number (26) in mammals, all named MED followed by a unique numerical designation (*Poss, Ebmeier & Taatjes, 2013*; *Allen & Taatjes, 2015*). In addition to this core Mediator complex, four additional subunits comprising the CDK8 or kinase module can associate with the core (*Poss, Ebmeier & Taatjes, 2013*). The Mediator complex likely co-evolved with basal transcription factors with a level of conservation between different phyla that is relatively low (*Poss, Ebmeier & Taatjes, 2013*; *Allen & Taatjes, 2015*). While most Mediator subunits are present in similar molar ratios and comprise the core complex, some subunits were found to be present in variable amounts when complexes were isolated from tissue culture cells (*Kulak et al., 2014*). Quantification of proteomically analyzed Mediator subunits showed that distinct forms of the complex have variable transcriptional activity (*Paoletti et al., 2006*) and analysis of Mediator complex subunits in *Drosophila* indicated that some subunits are critical only for specific gene transcription from endogenous genes but not for transcription from synthetic promoters (*Kim et al., 2004*). This diversity of Mediator subunit function reflects distinct transcription factor interactions with Mediator components and with Pol II, greatly expanding its possible regulatory roles for Mediator. For example, some Mediator subunits are essential for the transcription of many different protein-coding genes, while other subunits are essential for only a subset of genes, translating cellular signaling pathways to the regulation of specific target gene sets (reviewed in *Grants, Goh & Taubert (2015)*).

One of the Mediator complex subunits, MED28, is only found in higher eukaryotes. MED28 was originally identified as a gene expressed in endothelial cells where it was named EG-1 (Endothelial-derived Gene-1) (*Liu et al., 2002*); it was later shown to be part of the Mediator complex and re-named MED28 (*Sato et al., 2004*; *Beyer et al., 2007*). In addition to its role in the Mediator complex, the MED28 subunit has several cytoplasmic-associated interactions. MED28 has been shown to associate with (1) the actin cytoskeleton and linked to the regulation of smooth muscle genes (*Wiederhold et al., 2004*), (2) several Src-family kinases and it is a target of their phosphorylation (*Lee et al., 2006*), and (3) the plasma membrane where it interacts with Grb2 and Merlin (also called Neurofibromin 2 or Schwannomin) (*Wiederhold et al., 2004*), membrane-cytoskeleton scaffolding proteins linking actin filaments to the cell membrane (*McClatchey & Giovannini, 2005*; *McClatchey & Fehon, 2009*). These many and diverse cytoplasmic interactions suggest that MED28 could function to transmit cytoskeletal signals to transcription in the nucleus (*Lee et al., 2006*).

Although conserved between insects and mammals, a bona fide MED28 homologue had yet to be identified in nematodes. The relatively low conservation of Mediator complex subunits between eukaryotic phyla (*Poss, Ebmeier & Taatjes, 2013*; *Allen & Taatjes, 2015*) makes the identification of orthologues in distant species difficult and some suggested orthologues may require re-classification. Our previous work showed that the protein previously identified as "MDT-28" (**Med**ia**t**or-28) in nematodes (*Bourbon, 2008*) is instead the nematode homologue of perilipin, a protein regulating lipid metabolism at the level of lipid droplets and is not related to MED28 (*Chughtai et al., 2015*). Thinking it was unlikely that a MED28 homologue would be absent in nematode

genomes, we searched for it using the conserved features of MED28 orthologues from various phyla. Herein we identify a previously uncharacterized protein, F28F8.5, as the closest MED28 homologue. We show that F28F8.5 localizes to both nuclear and cytoplasmic compartments in most, if not all, cells throughout development. Downregulation by RNAi, or disruption of *F28F8.5* by deletion, results in multiple developmental defects during embryonic and larval development. Our work indicates that the homologue of Mediator complex subunit 28 exists in nematodes and suggests that the potential to link cytoplasmic and nuclear events is conserved between vertebrate and nematode MED28 homologues.

## MATERIALS AND METHODS

### Sequence analysis

The UniProtKB (http://www.uniprot.org) and NCBI (https://www.ncbi.nlm.nih.gov) databases were searched with BLAST, PSI-BLAST (*Altschul et al., 1997*), HHblits (*Remmert et al., 2011*), and HHpred (*Söding, Biegert & Lupas, 2005*) programs. The protein sequences were identified with their UniProtKB identifiers and the nucleotide sequences with their NCBI ones. The sequences were aligned with T-coffee (*Notredame, Higgins & Heringa, 2000*; *Di Tommaso et al., 2011*) and PROMALS (*Pei & Grishin, 2007*; *Pei et al., 2007*; *Pei, Kim & Grishin, 2008*). The secondary structure predictions were performed with PSIPRED (*Jones, 1999*; *Cuff & Barton, 2000*; *McGuffin, Bryson & Jones, 2000*). Multiple sequence alignments were displayed and analyzed with Jalview (*Clamp et al., 2004*).

### RNA isolation and cDNA synthesis

RNA and cDNA were prepared as described (*Zima et al., 2015*) with modifications. Cultured nematodes were collected in water and pelleted by centrifugation for 5 min at $200 \times g$ and 4 °C. The excess of water was removed and the pellet was frozen at −80 °C. For the isolation of RNA, the pellet was quickly melted and dissolved in 300 µl of resuspension buffer (10 mM Tris–HCl; 10 mM EDTA, 5% 2-mercaptoethanol; 0.5% SDS; pH 7.5). After adding 8 µl of proteinase K (20 mg/ml), the sample was mixed and incubated 1 h at 55 °C. RNA was isolated by phenol–chloroform extraction and ethanol precipitation. The obtained RNA was incubated with RQ1 DNase (Promega, Fitchburg, WI, USA) and purified again by phenol–chloroform extraction and ethanol precipitation. Complementary DNA (cDNA) was prepared with SuperScript III (Invitrogen, Carlsbad, CA, USA) using random hexamers.

### Strains, transgenic lines and genome editing

The *Caenorhabditis elegans* Bristol N2 strain was used whenever not specifically stated and maintained as described (*Brenner, 1974*).

KV3: (8418)—heterozygous animals carrying one edited disrupted allele of *F28F8.5* ($P_{F28F8.5}$ *(V:15573749)::gfp::let858(stop)::SEC::F28F8.5*—edited *F28F8.5* disrupted by *gfp* and self-excising cassette (SEC)) and one WT allele of *F28F8.5*. This line segregates mutant animals.

KV4: (8419)—edited *F28F8.5* carrying *gfp::F28F8.5* in its normal genomic position ($P_{F28F8.5}$(*V:15573749*)::*gfp::F28F8.5* on both alleles.

## Preparation of $P_{F28F8.5(400\ bp)}$::*F28F8.5::gfp*

For preparation of transgenic lines encoding F28F8.5::GFP from extrachromosomal arrays under regulation of endogenous promoter, we used the PCR fusion-based technique (*Hobert, 2002*). Primers 7886 and 7888 were used for amplification of the genomic sequence of *F28F8.5* (consisting of approximately 400 bp of the predicted promoter region preceding the coding region of *F28F8.5*). The gene encoding GFP was amplified from the pPD95.75 vector with primers 6232 and 6233. The complete construct was amplified with primers 7887 and 6234. The resulting fusion construct contains the 3′ UTR from pPD95.75 (originally from the *unc-54* gene). The PCR mixture was injected into the gonads of young adult hermaphrodite animals together with marker plasmid pRF4. The sequences of all primers used in the paper are in Supplemental Information.

## Genome editing

Lines with edited genomes were prepared from wild type N2 animals using the CRISPR/Cas9 system as described (*Dickinson et al., 2013, 2015*; *Ward, 2015*; *Dickinson & Goldstein, 2016*). Using this strategy, the *F28F8.5* gene was edited by insertion of a construct including the coding sequence of GFP and a SEC containing the *sqt-1(d)* gene (a visible selection marker leading to a Rol phenotype), *hs::Cre* (heat shock inducible Cre recombinase) and hygR (hygromycin resistance) genes. The sgRNA sequence was targeted near the start of the coding sequence for the *F28F8.5* gene using a modified pJW1219 plasmid (Addgene, Cambridge, MA, USA) as the Cas9 vector (pMA007); it was prepared by PCR with primers 8403A and 8333 and used in a concentration of 50 ng/µl for microinjections. The plasmid pMA007 was co-injected with the rescue repair template plasmid based upon modified pDD282 vector (pMA006) in a concentration of 10 ng/µl and with three markers (see below). The repair template plasmid pMA006 was prepared in two steps. First the plasmid pMA005 was prepared from gDNA of *F28F8.5* (containing both repair arms) and amplified by PCR with primers 8404 and 8405 and cloned into pCU19 backbone. The plasmid pMA005 was subsequently modified—the FP-SEC segment was added and the CRISPR/Cas9 site was altered to protect against Cas9 attack. The linear PCR product of pMA005 was prepared using primers 8406 and 8407 with overlapping regions for Gibson assembly (New England BioLabs, Ipswich, MA, USA). The primer 8406 was prepared with alternate codons for protection against CRISPR/Cas9 site. Linear insert of FP-SEC was prepared by PCR from pDD282 plasmid (Addgene, Cambridge, MA, USA) with primers 8408 and 8409. Primers were prepared with overlapping parts for cloning into linear pMA005 plasmid by Gibson assembly and the final rescue plasmid pMA006 was prepared. Plasmids pGH8 (10 ng/µl), pCFJ104 (5 ng/µl), and pCFJ90 (2.5 ng/µl) (Addgene, Cambridge, MA, USA) were used as fluorescent co-injection markers. After microinjections the population of nematodes were grown for three days at 25 °C and hygromycin (Invitrogen) was added in a final

concentration of 250 μg/ml. After three days integrated nematodes were selected according to the rolling phenotype and loss of extrachromosomal arrays.

Using this strategy, we obtained a heterozygous line (KV3) with a disrupted *F28F8.5* gene with an inserted *gfp* regulated by the endogenous promoter of *F28F8.5* in one allele and one WT allele. This line segregated homozygous animals for $P_{F28F8.5}$:: *F28F8.5::gfp* (edited *F28F8.5* with SEC—$P_{F28F8.5}$ *(V:15573749)::gfp::let858(stop)*::SEC::*F28F8.5*) with disrupted *F28F8.5* on both alleles and expressing GFP under the regulation of the endogenous promoter. Animals of this line were clearly distinguishable by their developmental phenotypes, weak expression of GFP in the cytoplasm and the presence of *rol* marker. These animals were sterile and had severe developmental defects (see Results). The genotypes were confirmed by single worm PCR of representative animals after their microscopic analysis (with primers 8398 and 8414).

The excision of the SEC was achieved by a 4 h heat shock at 34 °C. The line KV4 was obtained: animals with both alleles carrying the edited *F28F8.5* gene in the form of *gfp::F28F8.5* in its normal genomic position (edited *F28F8.5* with *gfp* tagged to the N—terminus—$P_{F28F8.5}$*(V:15573749)::gfp::F28F8.5*).

The presence of the knock-in of *gfp* was confirmed by single nematode PCR with primers 7887 and 8454, 8398 and 8454. The PCR products were purified and sequenced with primers 8455 and 8456. PCR was done by REDTaq ReadyMix PCR reaction (Sigma-Aldrich, St. Louis, Missouri, USA) or by Phusion High-Fidelity DNA Polymerase (New England Biolabs, Ipswich, MA, USA). During the maintenance of the heterozygous line KV3, animals with one edited disrupted *F28F8.5* allele and one allele with edited *F28F8.5* after self-excision of SEC were also generated (recognizable by the Rol phenotype, expression of *gfp* in nuclei and lack of developmental phenotypes). Schemes for genome editing are accessible in Files S4–S7.

## Downregulation of gene expression by RNA interference

For RNAi done by microinjections, *F28F8.5* cDNA was prepared from total cDNA using primers 7889 and 7890. The plasmid pPCRII(Topo) (Invitrogen, Carlsbad, CA, USA) containing *F28F8.5b* cDNA was linearized using restriction enzymes *Not*I/*Sac*I. The dsRNA was prepared by in vitro transcription using SP6/T7 Riboprobe® in vitro Transcription Systems (Promega, Madison, WI, USA) from opposing promoters synthesizing complementary single stranded RNA (ssRNA) for both strands of *F28F8.5* cDNA and its complementary strand. After in vitro transcription (~2 h) equal volumes of sense and antisense RNA were mixed, incubated at 75 °C for 10 min and slowly cooled to room temperature during 30 min. Control RNAi was prepared from the promoter region of *nhr-60* as previously described (*Šimečková et al., 2007*) and repeated with dsRNA prepared using the vector L4440. Vectors used for preparation of dsRNA were linearized and transcribed using T7 RNA polymerase. The dsRNA concentration was measured using a UV spectrophotometer and diluted to the concentration of ~2 μg/μl that was used for injections (*Tabara et al., 1999*; *Timmons, Court & Fire, 2001*; *Vohanka et al., 2010*).

## Microinjections

Microinjections of plasmids, DNA amplicons or dsRNA into gonads of young adult hermaphrodites were done using an Olympus IX70 microscope equipped with a Narishige microinjection system (Olympus, Tokyo, Japan). The plasmids were injected into the gonads of young adult hermaphrodites as described (*Tabara et al., 1999*; *Timmons, Court & Fire, 2001*; *Vohanka et al., 2010*).

## Microscopy

Fluorescence microscopy and Nomarski optics microscopy were done using an Olympus BX60 microscope equipped with DP30BW CD camera (Olympus, Tokyo, Japan). Animals were analyzed on microscopic glass slides with a thin layer of 2% agarose and immobilized by 1 mM levamisole (Sigma-Aldrich, St. Louis, MO, USA). Confocal microscopy of live homozygous animals with edited *F28F8.5* expressing *gfp::F28F8.5* was performed using an inverted Leica SP8 TCS SMD FLIM system equipped with a 63 × 1.2 NA water immersion objective, a pulsed white light laser (470–670 nm), AOBS and two internal hybrid single photon counting detectors, and operated by Leica Application Suite X program (Leica Microsystems, Wetzlar, Germany). The GFP fluorescence was excited at a wavelength of 488 nm and the emitted light was simultaneously recorded in two spectral ranges (Channel 1—495 nm to 525 nm, Channel 2—525 nm to 580 nm; the two channel setup was used to help resolve between spectrally different autofluorescence and GFP fluorescence signals).

## Fluorescence-lifetime imaging microscopy

For FLIM acquisitions the single photon counting signal from the internal hybrid detectors, acquired during confocal acquisitions, was simultaneously processed by HydraHarp400 TCSPC electronics (PicoQuant, Berlin, Germany) and information about the arrival times of all photons was stored to a hard-drive in TTTR data format. TTTR is freely accessible at https://www.picoquant.com/images/uploads/page/files/14528/technote_tttr.pdf. Data structure, program description and user instructions are also freely accessible at https://github.com/PicoQuant/PicoQuant-Time-Tagged-File-Format-Demos/blob/master/PTU/Matlab/Read_PTU.m. The signal from both time synchronized channels was added up. The false color scale (1–3 ns) is based on the average photon arrival time, with blue color representing short lifetime and red color long lifetime fluorescence.

## Single nematode PCR

Single animal PCR was used for verification of all transgenic lines. Following the microscopy examination, selected animals were removed from microscopic slides and transferred into caps of PCR tubes with 4 μl of solution of Proteinase K (20 mg/ml) diluted 1:333 in Barstead Buffer (resulting in Barstead Lysis Buffer which consists of 50 mM KCl, 10 mM Tris pH 8.3, 2.5 mM $MgCl_2$, 0.45% (v/v) NP40 (Nonidet P-40), 0.45% (v/v) Tween-20, 0.01% (w/v)). Proteinase K was diluted immediately before use as a 20 mg/ml stock solution which was kept on ice and diluted to final working solution

at a concentration of 60 μg/ml. The tube was sealed in bottom-up position and the sample transferred to the bottom of the tube by centrifugation. The tube was frozen for 10 min at −70 °C. Next, the tube was heated for 1 h at 65 °C and additional 15 min at 95 °C. The resulting sample was used immediately for amplification of DNA by PCR or stored at −80 °C before further analysis. Similarly, genomic DNA was prepared from selected nematode culture plates and used for further screening by PCR and sequencing.

The resulting precipitated DNA was dissolved in 10 μl of deionized water and used for amplification by PCR using primers outside the edited genomic regions. Specificity of amplification was confirmed by DNA sequencing.

Similarly, homozygous animals with edited *F28F8.5* (with *gfp* inserted in front of the *F28F8.5* START codon) were analyzed by single worm PCR with primers 7887 (sense primer) and 7890 or 8454 (antisense primers).

## Quantitative RT-PCR

For quantitative RT-PCR, the technique described by *Ly, Reid & Snell (2015)* was used with modifications. For assessment of the level of expression of *F28F8.5* from homozygous animals with the edited disrupted gene, five adult homozygous mutant animals recognized by the phenotype and the same number of young WT hermaphrodites with minimum number of formed embryos were manually harvested and collected in separate Eppendorf tubes. Reverse transcription was done using the Maxima H Minus cDNA synthesis kit (Thermo Fischer, Waltham, MA, USA) as recommended by manufacturer. Universal probe library and primers designed with the help of ProbeFinder Assay Design Software were used and qPCR was run on LightCycler 2.0 purchased from Roche (Roche, s.r.o. Prague, Czech Republic). An average of three sample cDNAs and three control cDNAs were analyzed (twice in duplicates and one time as single experiments), all containing the same amount of RNA for RT for each experiment. The expression of *F28F8.5* was normalized to *ama-1* and the values obtained in homozygous mutant animals with disrupted *F28F8.5* gene were compared to values obtained in control WT N2 animals.

## Binding studies

Binding studies were done as described (*Kostrouch et al., 2014*) with modifications. The coding region of *mdt-6* was amplified using primers 8292 and 8293 from cDNA prepared from mixed stages *C. elegans* cultures and cloned into pTNT vector (Promega, Madison, WI, USA, amplified with primers 8277 and 8278) using the Quick Ligation Kit (New England Biolabs, Ipswich, MA, USA) and expressed in the rabbit reticulocyte TNT-system (Promega, Madison, WI, USA). The in vitro transcribed protein was labeled using [35]S Methionine (Institute of Isotopes, Budapest, Hungary). F28F8.5 coding sequence (amplified using 8255 and 8256 primers with 15 bp overhangs for insertion into the vector) was cloned into pGEX-2T vector ((Amersham Pharmacia Biotech, Amsterdam, UK), amplified with primers 8253 and 8254) using the GeneArt Seamless PLUS Cloning and Assembly Kit (Thermo Fisher Scientific, Waltham, MA, USA), transformed into BL21 *Escherichia coli* cells and the production of protein was induced by

isopropyl $\beta$-D-1-thiogalactopyranoside (IPTG) (Sigma-Aldrich, St. Louis, MO, USA). The Mediator subunit MDT-30 was amplified from mixed stages *C. elegans* cDNA with sense primer 8302 and reverse primer 8527 (containing FLAG sequence), cloned into pET28a(+) vector ((Addgene, Cambridge, MA, USA), amplified with primers 8519 and 8520) using the Quick Ligation Kit (New England Biolabs), transformed into BL21 *E. coli* cells and induced by IPTG. The lysate from bacteria producing $His_6$-MDT-30-FLAG was used directly or purified on HiTrap Chelating HP column (GE Healthcare, Chicago, IL, USA). Proteins produced by the TNT system or bacterial lysates of bacteria transformed with FLAG labeled Mediator subunits were incubated with glutathione–agarose (Sigma-Aldrich, St. Louis, MO, USA) adsorbed with equal amounts of GST or GST-F28F8.5. Radioactively labeled proteins were detected using TRI-CARB 1600TR, Liquid Scintillation Analyzer (Packard, Meriden, CT, USA).

The resulting samples (labeled proteins bound to GST- or GST-F28F8.5) were separated by polyacrylamide gel electrophoresis. $^{35}$S-MDT-6 was visualized by autoradiography and subsequently, the gel containing radioactively labeled protein was localized using superimposed autoradiograms, excised and the radioactivity determined in the scintillation detector. FLAG-labeled MDT-30 was determined by Western blot using an anti-FLAG antibody (monoclonal anti-FLAG, M2 (Sigma-Aldrich)) and quantified densitometrically by ImageJ computer program (https://imagej.nih.gov/ij/download.html) (*Schneider, Rasband & Eliceiri, 2012*).

## RESULTS

### Identification of the closest homologue of vertebrate Mediator complex subunit 28 in *C. elegans*

To identify the *C. elegans* homologue of MED28, we queried protein databases with curated SwissProt sequences from UniProtKB. They comprised several mammalian and insect proteins (e.g., human MED28_HUMAN and *D. melanogaster* MED28_DROME). The more sensitive profile-to-profile HHblitz and HHpred algorithms provided hits to a *C. elegans* annotated protein F28F8.5a and b with highly significant *E*-values. According to Wormbase (WS248), two protein isoforms are produced from the *F28F8.5* gene, isoform a with the length of 200 amino acids and isoform b that has a two amino acid insertion at position 20 of the N-terminal evolutionarily non-conserved region. The best results were obtained when pre-aligned vertebrate and insect MED28 homologues were used as query in three iterations ($E < 10^{-48}$ and the probability of true positive >99.99%). When the pre-aligned nematode sequences homologues to F28F8.5 were used to query profiles of human or *Drosophila* sequences in reciprocal searches, MED28 proteins were obtained with equally significant scores. BLAST and PSI-BLAST searches in their standard settings were not able to reveal a significant hit ($E < 10^{-3}$); the only nematode hit was a *Trichinella spiralis* protein (E5RZQ1). However, when the searches in protein databases were limited to sequences from *Ecdysozoa* with *Insecta* excluded (conservative inclusion threshold $E < 10^{-6}$) in the first two iterations and then continued in the complete database of sequences from all species in the subsequent

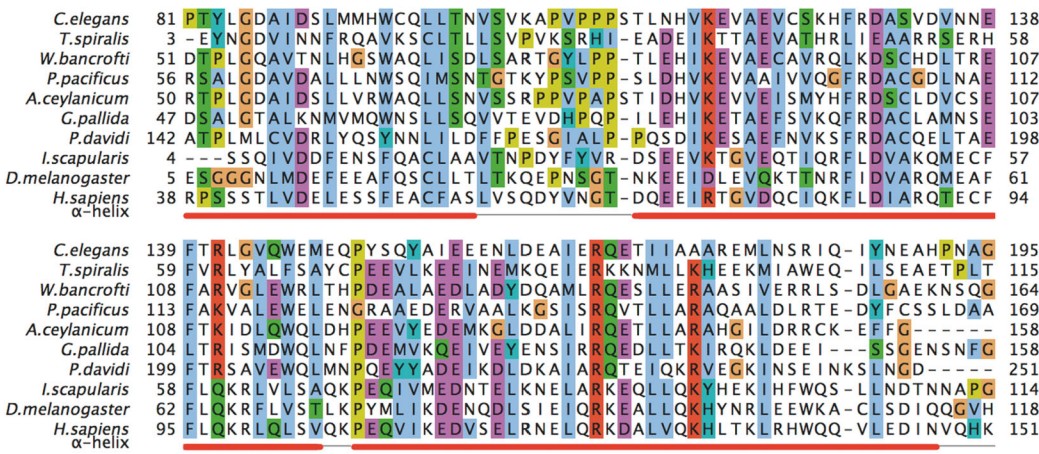

**Figure 1** **A multiple sequence alignment of selected metazoan homologues of MED28 compared with F28F8.5.** Aligned with PROMALS (http://prodata.swmed.edu/promals/promals.php), variable C- and N-termini not shown, amino acid residue types colored according to Clustal scheme in Jalview, red bars indicate consensus positions of predicted α-helices. Sequences from top to bottom (organism, identifier): *Caenorhabditis elegans*, O18692; *Trichinella spiralis*, E5RZQ1; *Wuchereria bancrofti*, EJW84794.1; *Pristionchus pacificus*, translated contig of CN657719.1 FG102945.1 CN657262.1 CN656622.1; *Ancylostoma ceylanicum*, A0A016SKV7; *Globodera pallida*, translated CV578368.1; *Panagrolaimus davidi*, translated JZ658977.1; *Ixodes scapularis*, B7PAW5; *Drosophila melanogaster*, MED28_DROME; *Homo sapiens*, MED28_HUMAN. Readers with specific color preferences may download the compared sequences from (File S1) and create the Clustal scheme with different color specifications using the Jalview program (http://www.jalview.org/).

iterations, the final hits of F28F8.5 included human and *Drosophila* MED28. PSI-BLAST with *T. spiralis* query sequence in database limited to *Ecdysozoa* in the first two iterations provided both human and *Drosophila* MED28 and F28F8.5 in one run ($E < 10^{-8}$). We concluded from these searches that F28F8.5 is a homologue of MED28 and very likely its previously unrecognized orthologue.

All PSI-BLAST MED28 homologues possess variable N- and C-termini of 3–80 amino acids showing no conservation across Metazoa. This conservation is loose even just within *Drosophilae* or *Caenorhabditae* sequences. Only the central core of about 110 amino acids is preserved in metazoan evolution. Figure 1 shows a sequence alignment of this conserved core of selected MED28 homologues. All sequences are predicted to fold into three helices forming a putative coiled coil fold (UniProt annotation). Submitting the alignment shown in Fig. 1 to HHpred for 3D structure recognition reveals a structural fold of yeast MED21 (PDB identifier 1ykh_B). It is indeed a three-helix coiled coil forming a heterodimer with MED7. It can be expected that MED28 forms a very similar fold interacting with a yet to be determined subunit of the MED complex.

## F28F8.5 is a nuclear as well as a cytoplasmic protein

Information available in WormBase suggests that the *F28F8.5* gene can be expressed as both an individual and multigene transcript, located as the last gene in a four gene operon that is both SL-1 and SL-2 trans-spliced. To determine the intracellular localization of F28F8.5, we edited the *F28F8.5* gene using CRISPR/Cas9 technology. We inserted the gene coding for GFP directly in front of the first codon. The arrangement used in our

experiment (based on *Dickinson et al. (2013, 2015)*, *Ward (2015)* and *Dickinson & Goldstein (2016)*) employed a SEC that was added after *gfp*. This strategy initially created a disrupted *F28F8.5* gene and putative null allele that can be detected by expression of GFP alone regulated by the endogenous promoter elements of *F28F8.5*. We found that only heterozygous animals could be propagated due to the sterility of homozygotes tagged in this manner. Assuming this tag is not deleterious to the expression of other genes in the operon, this result suggests that *F28F8.5* is an essential gene.

After removal of the SEC from this edited *F28F8.5* gene induced by heat shock (visualized by continuous expression of GFP::F28F8.5 fusion protein and loss of the Rol phenotypic marker), the endogenous locus had an N-terminus GFP-tagged *F28F8.5* gene that we maintained as homozygous animals, demonstrating this edited allele is fully functional. Note that both known protein isoforms of F28F8.5 (a and b) would be tagged on their N-terminus with GFP by this method.

The GFP::F28F8.5 pattern was ubiquitous, both nuclear and cytoplasmic from embryos to adults (Fig. 2). Prominent nuclear localization was found in oocytes, zygotes, larvae, and adults. Cells with clear nuclear accumulation of GFP::F28F8.5 included epidermal, intestinal, pharyngeal, uterine and vulval muscle cells (Fig. 2). The gonad expressed *gfp::F28F8.5* and mitotic as well as meiotic nuclei accumulated GFP::F28F8.5 protein (Fig. 2).

Selected animals were analyzed by confocal microscopy for determination of subcellular distribution of GFP::F28F8.5. Scanning through several focal planes revealed signal in the GFP excitation/emission range in nuclei as well as in the cytoplasm of embryos, all larval stages and adults (Fig. 3). Structures resembling gut granules were also strongly positive in the GFP recording mode. In order to distinguish between GFP-specific fluorescence and autofluorescence, we applied FLIM with an expectation that autofluorescence (such as that from gut granules) is likely to produce a signal with a short fluorescence lifetime opposed to GFP-specific fluorescence. Structures such as gut granules were clearly detected (Fig. 3, panels O, Q, S, T and U, blue color) while fluorescence with a longer lifetime expected for GFP::F28F8.5 was detected in the germline, in oocytes and embryos and in most somatic nuclei of larvae as well as adult animals (Fig. 3, panels O, Q, S, T and U, red and yellow colors).

We also generated transgenic lines encoding F28F8.5::GFP from extrachromosomal arrays consisting of an endogenous internal *F28F8.5* promoter regulating a fusion gene with *gfp* attached to F28F8.5 on its C-terminal end. As with the N-terminally tagged F28F8.5, F28F8.5::GFP showed both nuclear and cytoplasmic localization. As expected for an extrachromosomal transgene, the expression of *F28F8.5::gfp* was not detected in the germline. This reporter was expressed in embryos starting at the twofold stage and continued throughout development (Fig. S1). We did notice that *F28F8.5::gfp* was expressed in the excretory canal cell (Figs. S1M, S1N, S1P and S1Q), a pattern not observed with the endogenously edited GFP-tagged gene.

## F28F8.5 regulates development

To achieve loss-of-function, RNAi was used to downregulate *F28F8.5* expression. Analysis of 2,567 progeny of 17 young adult hermaphrodites inhibited for F28F8.5 function by

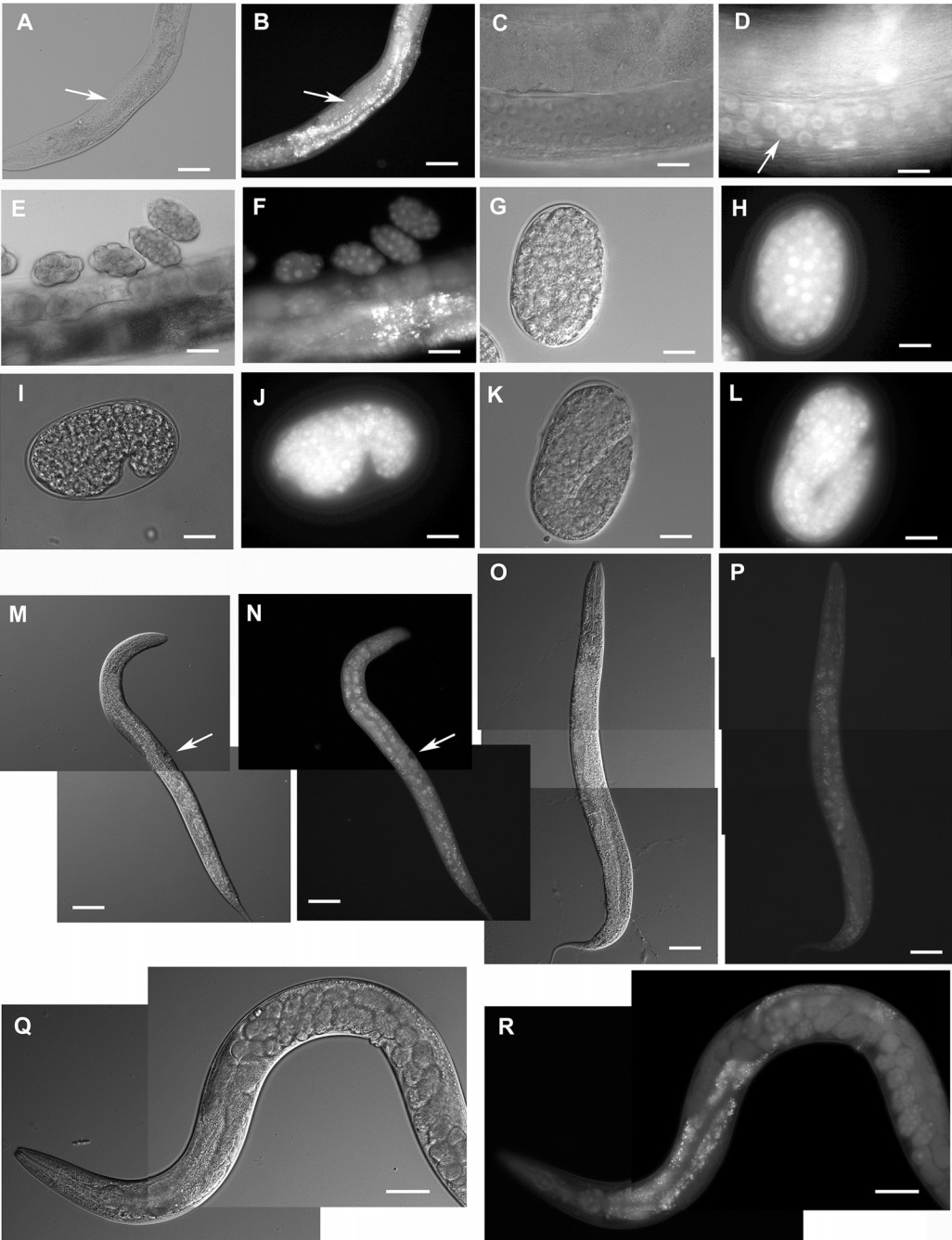

**Figure 2** **Expression pattern of GFP::F28F8.5 in homozygous animals with edited *F28F8.5* gene.** GFP tagged to F28F8.5 at its N-terminus using CRISPR/Cas9 technology visualized the expression of F28F8.5 in the gonads (Panels B and D, arrows) in mitotic nuclei and continues throughout the embryonic development (Panels F, H, J, and L). The wide and likely ubiquitous expression of GFP::F28F8.5 continues during larval stages (larvae L3 and L4 are shown in panels M, N and O, P, respectively) as well as in adults (panels Q and R). Expression of the edited gene in the nuclei of the developing vulva is indicated by the arrows in panels M and N. Panels A, C, E, G, I, K, M, O, and Q show larvae in Nomarski optics and panels B, D, F, H, J, L, N, P, and R in GFP fluorescence. Bars represent 50 μm.

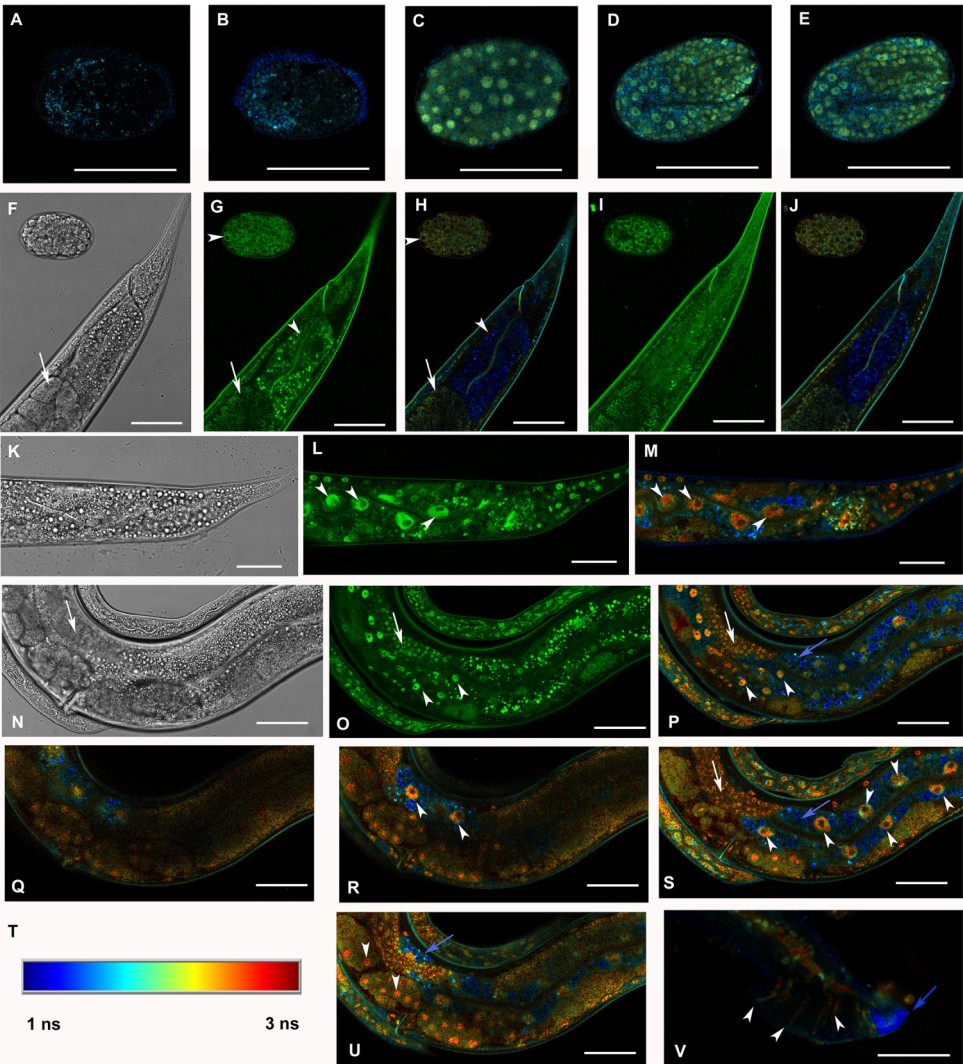

**Figure 3 Analysis of GFP::F28F8.5 expression in homozygous animals with edited *F28F8.5* gene by confocal microscopy and fluorescence lifetime imaging microscopy (FLIM).** All confocal images of GFP fluorescence are recorded in Channel 1 (495–525 nm). FLIM images (panels A to E, H, J, L, M, O to S, U, and V) are calculated from merged recordings in Channel 1 and Channel 2 (525–585 nm). Panels F, K, and N are images in Nomarski optics at the same optical focus as in corresponding confocal images of GFP fluorescence (panels G, I, L, O). Panels A and B show control images of WT embryos in bean and comma stages, respectively. No specific signal is detected in control embryos by FLIM. Panel C shows an embryo in bean stage expressing GFP::F28F8.5 from the edited gene. Two focal planes of an embryo expressing GFP::F28F8.5 in twofold stage are shown in panels D and E. FLIM detects GFP::F28F8.5 in most or all nuclei of developing embryos. Panels F to J show the distal part of a young adult control hermaphrodite animal and a control embryo in late bean stage. FLIM images in panels H and J show mostly short lifetime fluorescence in the cytoplasm of embryonic cells as well as cells and subcellular structures in the adult control animal (visualized by blue color). Arrowheads pointing at the embryo in panels G and H indicate weak autofluorescence in the cytoplasm of embryonic cells. Arrows in panels F, G and H indicate the turn of gonad and arrowheads indicate nuclei of an enterocyte which is devoid of almost all fluorescence (panels G and H). Panels K to S and U and V show animals with edited *F28F8.5* (*gfp::F28F8.5*). Panel M shows the distal part of an adult hermaphrodite animal expressing GFP::F28F8.5 from the edited gene at recording settings identical with that used in the control sample shown in panels A, B, H, and J. FLIM analysis shows a long lifetime fluorescence in nuclei and in the cytoplasm of most cells that contrasts with the low level of fluorescence seen in the control sample. Arrowheads indicate

**Figure 3** ... **continued**
nuclei of enterocytes in panels L and M. Panels N to V show images of an adult animal and two L1 larvae with edited *F28F8.5*. Panels P to S, and U show selected focal planes in FLIM. Panel T shows the calibration table for FLIM in the range of 1–3 ns used in all panels presenting FLIM analysis. Blue areas shown in FLIM pictures represent short lifetime fluorescence presumably corresponding to autofluorescence (blue arrows in panels O, S, and U). Arrowheads in panels O, P, R, and S indicate nuclei of enterocytes and in panel U nuclei of early embryos with long lifetime fluorescence characteristic for GFP. Panel V shows the distal part of a male expressing GFP:F28F8.5 in male specific structures, in nuclei as well as in rays (marked by arrowheads) indicating that GFP::F28F8.5 is expressed not only in cell nuclei but also in the cytoplasmic structures. Bars represent 30 μm in panels A to E and 50 μm in panels F to S and U and V.

microinjection of dsRNA into the syncytial gonad revealed that F28F8.5 is essential for proper development (Fig. 4). From the total progeny, 1,127 animals were affected (44%) exhibiting embryonic and larval arrest and a range of less severe phenotypes, including defective molting, protruding vulvae that often burst, male tail ray defects (Fig. 4), and uncoordinated (Unc) movement. In contrast with this, control young adult N2 hermaphrodites injected with control dsRNA showed embryonic arrest in less than 2% of progeny (seven hermaphrodites injected, total progeny observed 1,066, embryonic arrest found in 19 embryos).

Complete loss of F28F8.5 that occurred in homozygous animals with both edited disrupted alleles of the *F28F8.5* gene (that are found among the progeny of heterozygous animals carrying one edited disrupted allele and one WT allele) resulted in defective development that was most pronounced in late larval stages. The phenotypes included a dumpy phenotype (Dpy) (Fig. 5C), irregular gut, severely defective growth of the gonad with signs of defect in directional growth (Figs. 5E and 5M) and Pvul phenotype (Figs. 5K and 5L). Most animals had darker gut cells than controls of the same age. The gonad did not develop fully in most animals (Figs. 5C–5L) and often contained empty spaces that were prevalent in some animals, leading to the formation of larvae with optically thin, empty-like tissue in the position of the gonad. The gonads contained foci of irregular tissue with an uncharacteristic appearance. Tissue defects were also visible in extragonadal locations, especially in the place of excretory canals. Body defects were also observed in the position of the uterus that was not properly formed and the spermatheca that was not identifiable in a large proportion of animals.

Estimation of the level of *F28F8.5* expression in homozygous mutant animals (originating from maternal load or from SEC self-excision (*Dickinson et al., 2015*)) from three experiments indicated that mutants with disrupted F28F8.5 had the level of expression about 17 times lower compared to the levels found in WT controls (File S8).

Heterozygous hermaphrodites carrying one edited disrupted allele of *F28F8.5* and one WT allele were grossly normal and produced viable embryos. Unlike in homozygous animals carrying the excised SEC allele, the GFP fluorescence was mostly cytoplasmic and most nuclei were not showing accumulation of GFP. In some embryos, however, the nuclei accumulated GFP indicating probable spontaneous SEC self-excision (Fig. 6).

Analysis of progeny of the heterozygous strain KV3 revealed differences compared to the expected Mendelian segregation of phenotypes. Animals with one edited disrupted

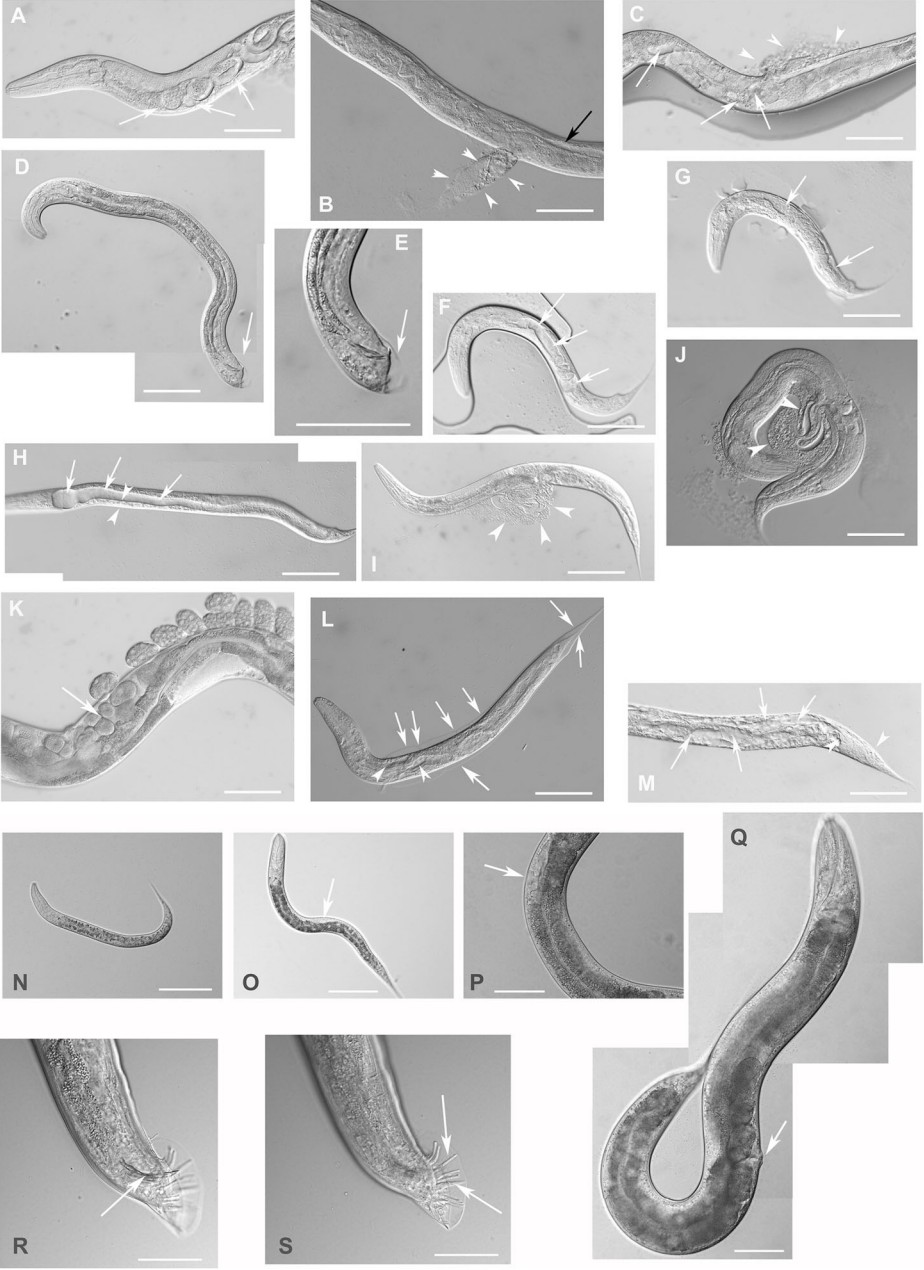

**Figure 4 Downregulation of *F28F8.5* by RNAi induces developmental defects.** Animals developed from parents injected with dsRNA specific for *F28F8.5* show retention of normal and malformed embryos (panels A and K, respectively), vacuoles (panels A and C, arrows), herniation and burst through the vulva (panels B, C, I, and J, arrowheads) and defective development of the gonad (panels J and M). Panel D shows a male nematode with defects of male specific structures—missing rays and fan and an abnormal distal part of the body (arrow). Panel E shows the magnified distal part of the male nematode in panel D and the defective male specific structures (arrow). Panels F and H show L3 larvae that were found atrophic, with thin enterocytes (arrowheads) and a dilated gut lumen (arrows). The dumpy phenotype with masses of tissue and vacuoles (panel G, arrows) were also common in the progeny of microinjected parents. Other phenotypes seen included molting defects indicated by arrows in panel L and cellular defects (indicated by arrowheads in panels L and M). Animals treated by control RNAi were morphologically normal and representative images are shown in panels N to Q. Panel N shows a L2

larva, panel O shows a young L3 larva with developing germline (arrow). Panel P shows a young L4 Larva with developing vulva marked by an arrow. Panel Q shows a grossly normal adult hermaphrodite animal with few developing embryos and vulva (arrow). Panel R and S show the distal part of the body of a male animal with normal appearance of male specific structures. Arrow marks spicules (in panel R) and normal sensory rays (in panel S). All images are in Nomarski optics. Bars represent 50 μm.

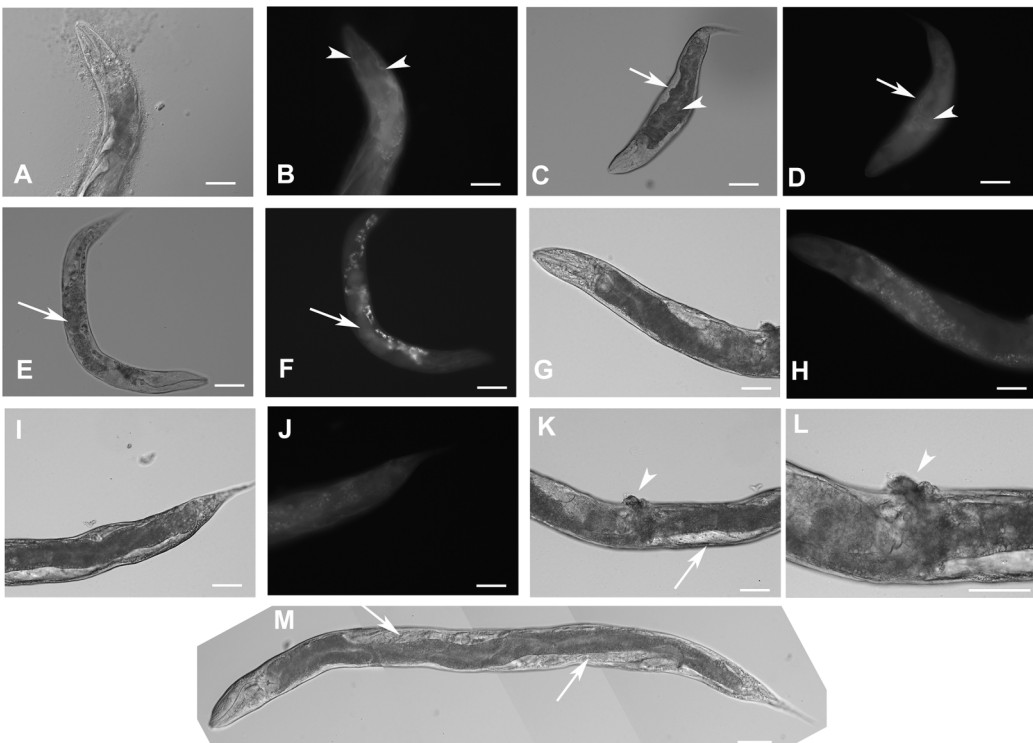

**Figure 5** **Disruption of *F28F8.5* by CRISPR/Cas9 technique.** Animals with disrupted *F28F8.5* on both alleles express GFP under the regulation of *F28F8.5* promoter. Panels A to J show paired images of animals in Nomarski optics and in GFP fluorescence. Panels A and B show an adult hermaphrodite animal with diffuse fluorescence in cells in the head area including anterior arms of the excretory cell (arrowheads). Panels C and D show a malformed larva probably in L3 stage with a Dpy phenotype and diffuse fluorescence in a malformed gonad (arrows) and the intestine (arrowheads). Panels E and F show an adult hermaphrodite animal with diffuse fluorescence in gut, pharyngeal cells and severely malformed gonad containing irregular structures (arrows). Panels G, H, I, and J show an adult animal with a malformed gonad, Pvul phenotype, dense gut and diffuse GFP fluorescence throughout the body. Panels K and L show the central part of the body of a hermaphrodite with the Pvul phenotype (arrowhead) and malformation of gonad (arrow). Panel M is composed of three consecutive images showing an adult hermaphrodite animal with severely malformed gonad (arrows), and missing uterus and spermathecae. The fluorescence images show that unlike GFP::F28F8.5, GFP alone localizes diffusely in the cytoplasm and is not found in nuclei. Bars represent 50 μm.

*F28F8.5* allele and one edited *F28F8.5* allele with excised SEC were detected. They were recognizable by the Rol phenotype, expression of GFP in nuclei and lack of developmental phenotypes. This genotype was supported by PCR amplification of genomic regions from single nematodes and obtained pattern of amplified DNA fragments. These lines were not stable and were not preserved.

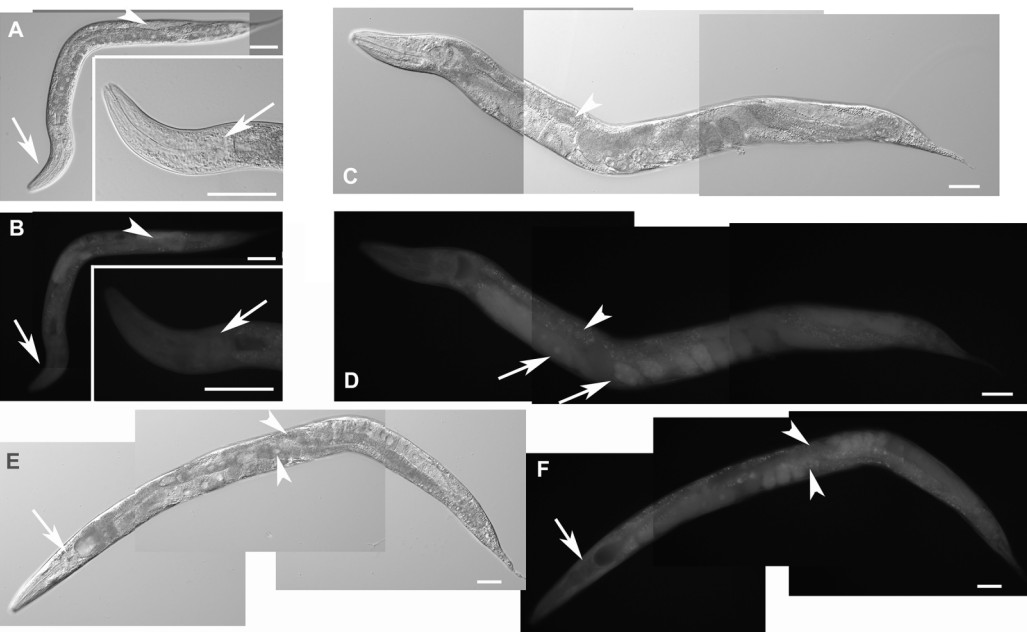

**Figure 6 Heterozygous animals with one edited disrupted allele of *F28F8.5* and one WT allele.**
Heterozygous hermaphrodites carrying one edited allele of *F28F8.5* and one WT had grossly normal appearance and could be recognized by *rol* phenotype, presence of embryos, weak mostly cytoplasmic GFP fluorescence and absence of nuclear localization of GFP fluorescence. Panels A (Nomarski optics) and B (GFP fluorescence) show an L3 larva with weak fluorescence (panel B, arrowhead points at the gonad and arrows point at the head and pharynx). Inlets show head area at higher magnification (rotated 90° clockwise). Panels C and D show an adult hermaphrodite animal (C in Nomarski optics and D in GFP fluorescence) with weak cytoplasmic fluorescence in most cells. The arrowhead in panel D points at the nucleus of an enterocyte in focal plane that is devoid of GFP fluorescence. Arrows indicate two embryos with GFP fluorescence accumulated in nuclei which is most likely the result of spontaneous SEC self-excision. Panels E and F show an adult hermaphrodite in Nomarski optics (panel E) and GFP fluorescence (panel F). Arrows indicate the head area with diffuse intracellular fluorescence visible in panel F. Arrowheads point at two nuclei of enterocytes in focal plane that are also devoid of fluorescence. In contrast to the animal shown in panels C and D, the animal shown in the panel E and F contains embryos that have mostly diffuse cytoplasmic expression of GFP. Bars represent 50 μm.

## F28F8.5 interacts with Mediator complex subunits

To determine if F28F8.5 could be part of the Mediator complex in *C. elegans*, we explored its ability to interact with previously identified Mediator subunits. We expressed $^{35}$S-labeled MDT-6, part of the "head" module where MED28 is located, in rabbit reticulocyte lysate and assayed its binding to bacterially expressed GST-F28F8.5 or to GST only. As shown in Figs. 7A and 7B, a strong interaction (~7.7-fold enrichment) was detected between MDT-6 to F28F8.5 that exceeded that seen with GST alone. We also assayed for interaction between GST-F28F8.5 and MDT-30, but we were unable to obtain a satisfactory $^{35}$S-Methionine labeled protein in the rabbit reticulocyte system. Therefore, we expressed MDT-30 containing a FLAG sequence inserted at the C-terminus and a His$_6$ sequence positioned at the N-terminus. After expression in bacteria and purification on a nickel column, we found that the MDT-30-FLAG bound F28F8.5 preferentially (~2.5-fold enrichment) in comparison to GST alone, as revealed by Western blot using an anti-FLAG antibody (Figs. 7C and 7D).
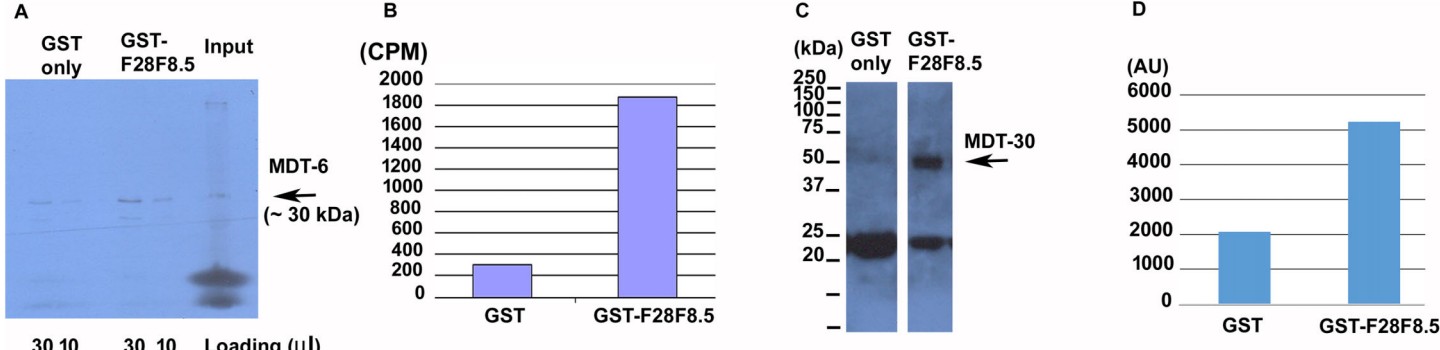

**Figure 7 Binding of F28F8.5 to MDT-6 and MDT-30 in vitro.** GST-F28F8.5 or GST alone were expressed in bacteria and purified using glutathione–agarose beads that were incubated with $^{35}$S-Methionine-MDT-6 produced using rabbit reticulocyte lysate (A and B) or His$_6$-MDT-30-FLAG expressed in bacteria and purified using a nickel column (C and D). Panels A and C show fractions bound to glutathione–agarose beads resolved by polyacrylamide gel electrophoresis and visualized by autoradiography (panel A). For quantification, dried gel areas corresponding to proteins detected by radioactivity were excised and the radioactively labeled MDT-6 was determined using scintillation counter (panel B). Panels C and D show the interaction of FLAG-MDT-30 with GST-F28F8.5 or GST alone. FLAG-MDT-30 pulled down by GST or GST-F28F8.5 was determined by Western blot using an anti-FLAG antibody (panel C) and by densitometry (panel D). Both assayed Mediator subunits, MDT-6 and MDT-30 bind GST-F28F8.5 preferentially in comparison to GST only.

# DISCUSSION

In this work, we identified an uncharacterized predicted protein F28F8.5 as the likely nematode homologue of MED28. This is supported by our findings that F28F8.5 interacts with nematode homologues of MED6 and MED30 (MDT-6, and MDT-30), the close sequence similarity of F28F8.5 to MED28 detected informatically in nematode genomes, a conserved dual nuclear and cytoplasmic expression pattern, and its involvement in a wide range of developmental processes. Thus we suggest F28F8.5 be identified as MDT-28, with the original gene for MDT-28 now recognized as the nematode homologue of perilipin, now named PLIN-1 (*Chughtai et al., 2015*).

The Mediator complex as a multiprotein structure is able to interact with a large number of additional proteins and integrate regulatory signals from several cell-signaling cascades (*Allen & Taatjes, 2015*). The Mediator complex possesses a high degree of structural flexibility and variable subunit composition (reviewed in *Poss, Ebmeier & Taatjes (2013)*). In yeast, a set of core subunits is important for a wide range of gene transcription (e.g., Med17 and Med22 (*Thompson & Young, 1995*; *Holstege et al., 1998*)), while others are non-essential in terms that single mutants can be maintained at laboratory conditions (Med1, Med2, Med3, Med5, Med9, Med15, Med16, Med18, Med19, Med20, Med31, and CDK module subunits Med12 (srb8), Med13 (srb9), srb10 (CDK8), and srb11 (CycC) (*Dettmann et al., 2010*)). Med31 and Srb9/Med13 Mediator subunits have different roles in gene selective transcription in *Saccharomyces cervisiae* and in *Candida albicans* (*Uwamahoro et al., 2012*). Mass spectrometry analyses indicate that many Mediator subunits are present in stoichiometric quantities but some subunits are over- or under-represented in budding and fission yeast and human HeLa cells (*Kulak et al., 2014*). Isolation of mammalian Mediator complexes lacking one or more of the 26 core subunits was reported for Med1 (TRAP220) (*Malik et al., 2004*), Med1 (Med220)

and Med26 (a.k.a. Med70 or CRSP70, or ARC70) (*Taatjes & Tjian, 2004*) (reviewed in *Allen & Taatjes (2015)*, the unified nomenclature can be found in (*Bourbon et al., 2004*) and for Med70 in *Rachez & Freedman (2001)*). While in cells with a stem cell like character, a broad spectrum of Mediator subunits was detected during; differentiation the expression of some Mediator subunits was markedly decreased (MED14, MED18, MED12, CDK8, MED26 in myotubes versus myoblasts) (*Deato et al., 2008*). In hepatocytes MED1, MED6, MED7, MED12, MED14, MED16, MED18, MED23, and CDK8 are decreased or even undetectable upon differentiation from hepatoblasts to hepatocytes (*D'Alessio et al., 2011*). Quantitative mass spectrometry analyses of Mediator complexes isolated by immunoprecipitation using four different Mediator subunits expressed as FLAG-tagged proteins in HeLa cells (Med10, Med26, Med28, Med29) identified most subunits of the Mediator complex in similar quantities with the exception of MED30, which was found in elevated ratios by Med28 pull-down experiments compared to other tested subunits. Med26 and Med29 precipitated more abundantly in their own pull-downs. Med31 was immunoprecipitated more efficiently in complexes with Med10 and Med25 was the least abundant subunit in all examined pull-downs (*Paoletti et al., 2006*). In-keeping with this, the abundance of individual Mediator subunits identified by quantitative proteomics indicate that some subunits are in similar abundance while others are under-represented or more numerous in yeast as well as in HeLa cells (*Kulak et al., 2014*). This suggests that Mediator complexes with specialized functions are likely to exist. A similar situation may be observed on ribosomes. Although the structure of ribosomal subunits is very firm and is given by the secondary structure of ribosomal RNAs and the presence of ribosomal proteins (*Ban et al., 2000*; *Schluenzen et al., 2000*; *Wimberly et al., 2000*), ribosome function during translation of mRNAs can be effectively regulated by viral proteins (*Diaz et al., 1993*, *1996*) which reveals the existence of a regulable "ribosomal code." The regulation of ribosome biogenesis and translation through the p53 pathway and methylation of ribosomal RNA by fibrillarin is leading to cancer specific ribosomes (*Marcel et al., 2013*). Cells infected with the viral oncogene v-erbA, the viral form of thyroid hormone receptor alpha, produce ribosomes with decreased levels of RPL11 which are translating more effectively Hsp70, a protein critical for tumorigenesis in avian erythroblastosis (*Nguyen-Lefebvre et al., 2014*). In comparison to ribosomal subunits, the Mediator complex possesses some analogies and differences.

The unit that is forming the structural backbone of Mediator is MED14 which is critical for both basal and activated transcription (*Cevher et al., 2014*). Mediator complexes bound to specific transcription factors (SREB Mediator, VP16-Mediator, TR-Mediator, VDR-Mediator, p53-Mediator) and the unliganded Mediator assume all distinct sterical conformations with fundamentally altered exposed protein surfaces (*Poss, Ebmeier & Taatjes, 2013*) that can be expected to form a multipotent basis for additional protein-protein interactions. This is possible because the fundamental features of the Mediator subunits are their intrinsically disordered regions that are to a certain degree positionally conserved between species, while others evolved in a phylum or species-specific way (*Nagulapalli et al., 2016*). In yeast, Med3 and Med15 form amyloid-like protein aggregates under $H_2O_2$ stress conditions. The amyloid formation can be

induced by overexpression of Med3 or glutamine-rich domain of Med15. This subsequently leads to the loss of Med15 module from Mediator and a change in the stress response (*Zhu et al., 2015*). The Mediator complexes contact a wide range of transcription factors using a fuzzy protein interface (*Brzovic et al., 2011*; *Warfield et al., 2014*). It can be therefore anticipated that additional proteins with a similar protein–protein interaction potential have the capability to interact with Mediator subunits if they are translocated into the nucleus.

Although individual Mediator complex subunits were shown to be associated with specific functions (reviewed in *Grants, Goh & Taubert (2015)*), the function of the nematode orthologue of MED28 could not be studied since it was not yet identified. MED28 has a special position in-between Mediator subunit proteins for its dual regulatory role, one as a Mediator subunit (*Sato et al., 2004*; *Beyer et al., 2007*) and the second, which is cytoplasmic, at the level of the cytoskeleton (*Wiederhold et al., 2004*; *Lee et al., 2006*; *Lu et al., 2006*; *Huang et al., 2012*). It can be anticipated that the interaction of primarily cytoplasmic proteins with MED28 if translocated to the nucleus may be able to bring cytoplasmic regulatory interactions towards the regulation of gene expression. In-between cytoplasmic proteins regulating gene expression, probably the most studied is beta-catenin, an adaptor of interaction between the cytoskeleton and cell adhesion molecules which critically regulates gene expression in the Wnt pathway. This connection is known in *C. elegans* to great detail (reviewed in *Grants, Goh & Taubert (2015)*). Interestingly, the phenotypes that we observed in F28F8.5 knock-down and loss of function experiments overlap with the EGFR regulatory cascade in *C. elegans*, especially the developmental defects of the vulva and of male specific structures, most obviously, male rays (*Grants, Goh & Taubert, 2015*; *Grants et al., 2016*). Our observation of the expression of *F28F8.5* in male rays and the defective development of male specific structures after *F28F8.5* RNAi support the cytoplasmic role of F28F8.5, that is in mammals mediated by Grb2 (*Wiederhold et al., 2004*). This cytoplasmic function of F28F8.5 is supported by the known involvement of the nematode homologue of Grb2, SEM-5, in the regulation of development of male rays. F28F8.5 protein contains a predicted SH2 binding site for Grb2 in the loop positioned in-between the two helices of F28F8.5, similarly as MED28 (identified using the site prediction tool Motif Scan http://scansite.mit.edu/motifscan_seq.phtml) (*Wiederhold et al., 2004*). Although, it has to be stressed that there are no close structures available for a high-probability prediction of the structure of F28F8.5. The burst through vulva phenotype is also likely to be connected to LET-60/Ras signaling (*Ecsedi, Rausch & Grosshans, 2015*) that also supports the conservation of the dual, nuclear and cytoplasmic functions, of MED28 homologues throughout the evolution of Metazoa.

We propose that MED28 is a candidate Mediator complex subunit linking cytoplasmic structural signals towards the core of transcription regulation. The connection between cytoplasmic events and regulation of gene expression can be seen frequently. Numerous transcription factors are regulated by their spatial restriction, binding or incorporation into cytoplasmic structures and organelles. Many proteins that have primarily cytoplasmic structural functions were shown to possess transcription regulating activity (e.g., proteins

interacting with steroid receptors (*George, Schiltz & Hager, 2009*), FOX transcription factors (*Gan et al., 2005*; *Wang et al., 2015*), and BIR-1/Survivin (*Kostrouch et al., 2014*)). In-between interactions of MDT-28 that we identified, the interaction with MDT-30 may suggest an additional link towards connection of structural signals with the regulation of gene expression. MED30 was shown to be pulled down by MED28 quantitatively with higher efficiency compared to other subunits, possibly suggesting that these two subunits may be present in some subpopulations of Mediator complexes that could lack other Mediator subunits. MED30 is similarly as MED28 a likely more recent Mediator subunit specific to Metazoa and absent in yeast and it is intriguing to speculate that the more recently evolved Mediator subunits are linked with the evolution of structurally differentiated cells and tissues. It can be anticipated that impairment of cellular structure sensing could be involved in cancer biology. In-keeping with this, MED30 was recently identified as an upregulated gene in stomach cancer connected with cancer proliferative properties (*Lee et al., 2015*) and in development of cardiomyopathy in mice carrying a missense mutation in the first exon (*Krebs et al., 2011*). MED28 was also connected with cancer behavior and migration of cancer cells (*Huang et al., 2012*, *2017*). Wormbase also lists phenotypes similar to F28F8.5 knock-down by RNAi for *mdt-30*, namely a Dpy, burst through vulva and locomotion defect but not a germline defect (Wormbase WS, accessed on March 11, 2017). The gene *mdt-30* is organized in an operon together with F44B9.8 which is an ortholog of human RFC5 (replication factor C subunit 5) and its inhibition by RNAi leads to embryonic defects. Similarly as F28F8.5, *mdt-30* is likely to be expressed independently from the operon since it is trans-spliced with both SL1 and SL2 splice leaders (Wormbase WS, accessed on March 11, 2017).

Our results demonstrated phenotypic differences comparing knock down versus knockout of F28F8.5 activity. For example, downregulation of *F28F8.5* by RNAi resulted in embryonic lethality and larval arrest whereas null mutants with a disrupted *F28F8.5* gene found in the progeny of heterozygous animals with one edited disrupted allele and one WT allele or one edited disrupted allele and one edited allele coding for GFP::F28F8.5 were able to reach adulthood. Morevoer, most phenotypes that we observed in our RNAi experiments have previously been reported in Wormbase (WS254) based on high throughput screens (*Kamath & Ahringer, 2003*; *Simmer et al., 2003*; *Frand, Russel & Ruvkun, 2005*; *Sönnichsen et al., 2005*). One explanation of the differences between knockdowns versus knockouts is that heterozygous animals with one functional allele of *F28F8.5* supply their embryos with maternal transcripts, while the embryos in the progeny of parents with *F28F8.5* downregulated by RNAi are devoid of this maternal load; maternal rescue of loss-of-function mutations is frequently observed in *C. elegans* early development. This model further predicts that the amount of *F28F8.5* product inherited maternally is not sufficient for normal development of the gonad and other post-embryonic developmental events such as male tail development. Alternatively, our F28F8.5 disruptions could be affecting other genes in the operon, although none have been reported to result in high level embryonic lethality when eliminated individually. The three other genes within this operon are *atx-3*, the orthologue of human ataxin-3,

F28F8.9, a non-characteristic predicted protein and F28F8.7, an orthologue of human ELMSAN1 (ELM2 and Myb/SANT domain containing 1) and TRERF1 (transcriptional regulating factor 1). RNAi experiments have been reported for *atx-3* and F28F8.9, of which only inhibition of *atx-3* produced embryonic arrest in 10–25% of embryos. This suggests that even if our gene disruption is affecting other genes in the operon (as reported in *Bosher et al. (1999)*), the severe larval changes reported here are most likely the consequence of inhibition of *F28F8.5*. In addition, *F28F8.5* is also expressed independently from its own promoter based on our translational fusions and reported SL1 splice leader (*Mounsey, Bauer & Hope, 2002*; *Matus et al., 2010*). Further studies will be required to sort out the potentially complex interactions among these genes in development.

The broad expression pattern and indispensability of F28F8.5 we find during embryonic development is similar to findings reported for Med28 in other systems ((*Li et al., 2015*); Mouse Genome Database (http://www.mousephenotype.org/data/genes/MGI:1914249) (*Eppig et al., 2015*); Human Protein Atlas (http://www.proteinatlas.org) (*Uhlén et al., 2015*)). F28F8.5 was also shown to have tissue-specific functions, as in the anchor cell where it is important for the regulation of anchor cell translocation across the basement membrane during the formation of the developing vulva (*Matus et al., 2010*).

Our experiments with transgenes fused to GFP also show the differences between the expression of N- or C-terminally labeled F28F8.5. The expression of fusion transgenes is not entirely without functional and developmental consequences. N-terminally labeled F28F8.5 is likely to be able to maintain the nuclear functions of F28F8.5. It is also able to support, at least partially the cytoplasmic functions of F28F8.5 in male rays, since they are formed but are not entirely normal and defects in some animals were observed. It seems likely that GFP-labeled subunits in viable lines may help localize the place of action of labeled proteins as well as their function in *C. elegans*. We did not observe elevated cytoplasmic expression of GFP::F28F8.5 in the anchor cell described by *Matus et al. (2010)*. This is probably dependent on the position of GFP in the fusion protein which was on the C-terminus, in the case of the study by *Matus et al. (2010)* as well as in the case of the data reported by Wormbase (both based on a clone originally prepared by Ian Hope). It has been suggested that proteins containing GFP at their C-terminus are more frequently properly intracellularly localized compared to proteins containing GFP at their N-terminus (*Palmer & Freeman, 2004*). GFP positioned at the N-terminus might also fold differently and fail to produce fluorescence in oxidizing compartments (*Aronson, Costantini & Snapp, 2011*).

The direct link between effector proteins and the regulation of transcription can be traced to Eubacteria and Archaea. Lrp/AsnC proteins, metabolic effectors in Archaea and related Lrs14 proteins are serving as multipotent (Lrp) and specific (Asn) regulators of gene expression. Lrs14 has a clear negative autoregulatory potential illustrating the ancient origin of the transcriptional function of effector proteins (*Bell & Jackson, 2000*; *Thaw et al., 2006*; *Orell et al., 2013*). Similarities between the core transcriptional machinery of Eukaryotes and Archaea can be clearly found (*Hirata & Murakami, 2009*).

While the archaeal transcriptional complex seems to be sufficiently dependent on two basal transcriptional regulators, TBP and TFB, Pol II dependent transcription in higher eukaryotes requires five or six general transcription factors (reviewed in *Burton et al. (2016)*) and the modular assembly of the Mediator complex at the promoters of regulated genes. This modular complex is capable of linking the informatic network necessary for cells differentiated to multiple cell types (or in other words multiple structural cell states) with gene expression. MED28 homologues are thus likely to be able to bring cytoplasmic proteins to the core of gene transcription. This may explain why MED28 evolved in multicellular eukaryotes containing structurally differentiated cells.

In conclusion, MED28 homologues in vertebrates, insects, and nematodes share similarities indicating their conserved roles in cytoplasmic and nuclear events. It can be hypothesized that many proteins that are primarily building blocks of cellular structures and structure-associated proteins are likely to be part of regulatory loops that regulate gene expression. Similarly, as is the case of evolution of operons in Rhabditida that are formed during evolution if they are biologically tolerated for the sake of other regulatory or energetic gains (*Qian & Zhang, 2008*; *Blumenthal, 2012*), regulation by structural proteins may also be evolving for a limited number of structural proteins leaving other structure-forming proteins available for evolution of other functions. The homologues of MED28 in mammals, insects and nematodes therefore may be a link between cellular structural states and regulation of gene expression.

## ABBREVIATIONS

[1] *F28F8.5* is now renamed with WormBase approval to *mdt-28*.

| | |
|---|---|
| **F28F8.5**[1] | gene coding for protein F28F8.5 |
| **F28F8.5a** | splice form a |
| **F28F8.5a** | protein form a |
| **F28F8.5b** | splice form b |
| **F28F8.5b** | protein form b |
| **gDNA** | genomic DNA |
| $\mathbf{P_{F28F8.5}(V:15573749)::}$ **gfp::F28F8.5** | edited *F28F8.5* with *gfp* tagged to the N-terminus in the position *V:15573749* (allele named edited *gfp::F28F8.5*) |
| $\mathbf{P_{F28F8.5}\ (V:15573749)::}$ **gfp::let858(stop):: SEC::F28F8.5** | edited *F28F8.5* disrupted by *gfp* and SEC (allele named edited disrupted *F28F8.5*) |
| $\mathbf{P_{F28F8.5(400\ bp)}::}$ **F28F8.5::gfp** | *F28F8.5* tagged with *gfp* on its C-terminus regulated by its predicted internal promoter with the size of 400 bp upstream of the ATG |
| **GFP::F28F8.5** | protein F28F8.5 tagged on its N-terminus with GFP |
| **F28F8.5::GFP** | protein F28F8.5 tagged on its C-terminus with GFP |

| *MED28* | vertebrate Mediator complex subunit 28 gene |
| **MED28** | vertebrate Mediator complex subunit 28 protein |
| **Med28** | Mediator complex subunit 28 in a general sense; this nomenclature is also used in *Drosophila* and mouse gene nomenclature |

## ACKNOWLEDGEMENTS

The authors thank WormBase and NCBI for accessibility of data and bioinformatic support and Caenorhabditis Genetics Center CGC for the N2 wild type strain. The authors thank Dr. Heather Etchevers, Dr. Stefan Taubert and the two anonymous reviewers for their very helpful comments, suggestions and corrections.

### Funding

This work was supported by the European Regional Development Fund "BIOCEV—Biotechnology and Biomedicine Centre of the Academy of Sciences and Charles University in Vestec" (CZ.1.05/1.1.00/02.0109) (The Start-Up Grant to the group Structure and Function of Cells in Their Normal State and in Pathology—Integrative Biology and Pathology (5.1.10)) and the LQ1604 National Sustainability Program II (Project BIOCEV-FAR) and the project Biocev (CZ.1.05/1.1.00/02.0109) from the Ministry of Education, Youth and Sports of Czech Republic; the grant PRVOUK-P27 and PROGRES Q26/LF1 from Charles University in Prague; the grant SVV 260377/2017, SVV 260257/2016, SVV 260149/2015 and SVV 260023/2014 from Charles University in Prague. This work was supported by the project OPPK No. CZ.2.16/3.1.00/24024, awarded by European Fund for Regional Development (Prague & EU—We invest for your future). MWK is supported by the Intramural Research Program of the National Institute of Diabetes and Digestive and Kidney Diseases (NIDDK) of the National Institutes of Health, USA. This work was supported in part by a monetary gift from MediCentrum Praha, Czech Republic. The imaging was done at the Imaging Methods Core Facility at BIOCEV, supported by the Czech-BioImaging large RI project (LM2015062 funded by MEYS CR). ZK and MK contributed with personal funds to this work. The funders (except authors) had no role in study design, data collection and analysis, decision to publish, or preparation of the manuscript.

### Grant Disclosures

The following grant information was disclosed by the authors:
European Regional Development Fund "BIOCEV—Biotechnology and Biomedicine Centre of the Academy of Sciences and Charles University in Vestec":
CZ.1.05/1.1.00/02.0109.
The Start-Up Grant: 5.1.10.
Ministry of Education, Youth and Sports of Czech Republic: BIOCEV-FAR, CZ.1.05/1.1.00/02.0109.

Charles University in Prague: PRVOUK-P27, PROGRES Q26/LF1, SVV 260377/2017, SVV 260257/2016, SVV 260149/2015 and SVV 260023/2014.
European Fund for Regional Development: CZ.2.16/3.1.00/24024.
Intramural Research Program of the National Institute of Diabetes and Digestive and Kidney Diseases (NIDDK) of the National Institutes of Health, USA.
MediCentrum Praha, Czech Republic.
Czech-BioImaging large RI project (MEYS CR): LM2015062.

## Competing Interests

Marta Kostrouchová is an Academic Editor for PeerJ. No other competing interests declared.

## Author Contributions

- Markéta Kostrouchová conceived and designed the experiments, performed the experiments, analyzed the data, wrote the paper, prepared figures and/or tables, reviewed drafts of the paper.
- David Kostrouch conceived and designed the experiments, performed the experiments, analyzed the data, wrote the paper, prepared figures and/or tables, reviewed drafts of the paper.
- Ahmed A. Chughtai conceived and designed the experiments, performed the experiments, analyzed the data, wrote the paper, prepared figures and/or tables, reviewed drafts of the paper.
- Filip Kaššák conceived and designed the experiments, performed the experiments, analyzed the data, wrote the paper, prepared figures and/or tables, reviewed drafts of the paper.
- Jan P. Novotný conceived and designed the experiments, performed the experiments, analyzed the data, wrote the paper, prepared figures and/or tables, reviewed drafts of the paper.
- Veronika Kostrouchová performed the experiments, analyzed the data, wrote the paper, prepared figures and/or tables, reviewed drafts of the paper.
- Aleš Benda conceived and designed the experiments, performed the experiments, analyzed the data, wrote the paper, prepared figures and/or tables, reviewed drafts of the paper.
- Michael W. Krause conceived and designed the experiments, performed the experiments, analyzed the data, wrote the paper, prepared figures and/or tables, reviewed drafts of the paper.
- Vladimír Saudek conceived and designed the experiments, performed the experiments, analyzed the data, wrote the paper, prepared figures and/or tables, reviewed drafts of the paper.
- Marta Kostrouchová conceived and designed the experiments, performed the experiments, analyzed the data, contributed reagents/materials/analysis tools, wrote the paper, prepared figures and/or tables, reviewed drafts of the paper, contributed with personal funds.

- Zdeněk Kostrouch conceived and designed the experiments, performed the experiments, analyzed the data, contributed reagents/materials/analysis tools, wrote the paper, prepared figures and/or tables, reviewed drafts of the paper, contributed with personal funds.

## Data Availability

The raw data has been supplied as Supplemental Dataset Files.

## Supplemental Information

Supplemental information for this article can be found online at http://dx.doi.org/10.7717/peerj.3390#supplemental-information.

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
