# Peer review of "The nematode homologue of Mediator complex subunit 28, F28F8.5, is a critical regulator of C. elegans development"

_PeerJ, doi:10.7717/peerj.3390_

## Round 0.1 · original submission · Major Revisions

I have requested the opinions of three expert reviewers and thank you for your patience in awaiting their commentary. All of them seem to find that the work is worthwhile and would benefit from a number of detailed improvements, which each suggests. There are some objections to interpretation of context and results relative to orthology, and your response to these objections should be detailed and documented with additional functional data, if possible. Given the general policy of this journal, I support making the code available for the TTTR data analysis. We look forward to reading your revised manuscript.

·

Basic reporting

The manuscript is overall well written and easy to read. However, I disagree somewhat with the content of the introduction and discussion as it pertains to Mediator function. Additionally, the prior annotation of mdt-28 gene, as done by Bourbon 2008, should be discussed in detail in the introduction, as it is relevant here. I have three major comments:
1) I don’t think that Mediator function as described here reflects the work of the cited papers, and the current opinion on Mediator function. The authors write "The steric organization of the Mediator complex is dynamic and allows variable arrangement of its subunits (Poss et al. 2013). Individual subunits of the Mediator complex can be divided into essential subunits, present in many or perhaps all Mediator complexes and proteins that are present only in some complexes and participate in specific more restricted transcription regulation (Paoletti et al. 2006; Kulak et al. 2014).” I think this contains some imprecisions. As far as I know, there is no “clear division” into essential subunits that occur in all Mediator complexes, and non-essential subunits that occur only in a few. From most available data in the species wherein it has been studied, it appears that Mediator subunit composition is highly similar and doesn’t change much depending on the method of purification (quote from the Paoletti paper cited here: "there is little significant difference in the relative stoichiometry of most Mediator subunit complexes purified through any of four different baits, Med26, Med10, Med28, and Med29”). In some cases, paralogs can substitute for each other (e.g. CDK8 and CDK19 in mammals), but the complex is always thought to contain ONE of these paralogs. The one change that the Paoletti paper described, and that has also been observed in other studies, is that there are two variations of Mediator: one seems to contain, on average, more Med26 and less CDK module and is highly active; the other is the opposite. However, in this represents an active-inactive distribution, and has nothing to do with functional diversity and specificity. Instead, the diversity of Mediator function appears to derive not from changing subunit composition but rather from its highly flexible structure that allows it to adopt different confirmations and hence to perform different activities depending on what protein interactions it undergoes. Poss et al (2013) discuss this in great detail.
2) Two key omissions from the citation include the Bourbon paper noted above, which identified MDT-28. I also found it surprising that the authors failed to cite the recent review on C. elegans Mediator by Grants et al in NAR, 2015, which discusses in detail the roles of complex subunits known to date. Please add this reference.
3) Integration with published work could be improved. The gene hasn’t been characterized in detail as done by this paper before, but it has been subject to study in numerous large scale studies. These have identified RNAi phenotypes, GFP expression patterns, etc. This is noted in the discussion but should probably be mentioned earlier. the authors should state clearly whether the phenotypes they observe agree with the ones found in large scale screens - if yes, it supports their study.

Experimental design

Overall, the experiments are generally well done and controlled. I have four comments:
1) Line 338: I’m not clear how the "endogenous internal F28F8.5 promoter" was identified - does this sequence have any elements that suggest it is a promoter? Or any ChIP signals from modENCODE that show the Pol2 binds? If not, how was it chosen? Can we be certain this is a promoter? Previously, it was discussed that this gene is a part of an operon. So why is there an internal promoter in F28F8.5? If this cannot be satisfyingly cleared up I’d suggest removing the data from the manuscript.
2) Line 347 and below: the knockdown and, ideally, the knockout of F28F8.5 needs to be validated. Specifically, the authors should perform qPCR to assure that F28F8.5 expression was actually rescued following the RNAi treatment, and ideally, also in the KO worms; while this is a bit more challenging due to the essential nature of the gene, the authors themselves state that "these animals were able to develop to adults”. It should be possible to pick a large enough number of homozygote worms and perform qPCR on them to ascertain reduced gene expression.
3) Figure legend of Fig 2 "The expression of the edited gene is relatively faint and panels B, D, F, H, J and L are shown after contrast enhancement by the Auto Contrast tool of Adobe Photoshop program 7.0”. This is a bit unusual, are the authors certain that this introduces no bias or artefact
4) Line 244: the software described as "home written software "TTTR data analysis” should be described in details, code potentially added in supplements or made available upon request

Validity of the findings

Kostrouchova et al describe the putative identification of F28F8.5 as Mediator subunit MED28 of C. elegans. They us bioinformatic analysis to show that this gene is related to MED28 subunits from other species, and perform expression and function analysis in vivo using GFP tagging and RNAi and knockout studies, finding broad expression and functional requirements in embryonic and larval development. The bona fide identification of this gene as a Mediator subunit would certainly be interesting to investigators in the field, including myself. Alas, I don't think that the current data provide convincing, conclusive evidence that this gene indeed encodes a bona fide MED28 ortholog.
Previously, Bourbon identified MDT-28 (W01A8.1, now renamed PLIN-1 following a rather convincing paper from these same authors suggesting that this gene encode a perilipin rather than Mediator subunit) as the MED28 ortholog, using computational predictions. In this paper, the authors also use computational prediction to ascertain that F28F8.5 is the MED28 ortholog. I’m no computational biologist but it seems as though they had to push the systems quite a bit to get strong alignments, as stated e.g. in "The best results were obtained when pre-aligned vertebrate and insect MED28 paralogues were used as query in 3 iterations”. Visually, alignment in Fig 1 look unconvincing between the human and the C. elegans protein. I'm not more convinced that F28F8.5 is the MED28 ortholog than I was that W01A8.1 is the MED28 ortholog.
All other experiments are not specific to MED28 - i.e., they provide no evidence that the gene studied MUST be a MED28 ortholog. The experiments are well done, but they simply show that the gene is widely expressed, sometimes nuclear, and has broad functions. Molecularly, it could be doing anything. In the absence of any molecular evidence that F28F8.5 indeed is a MED28 ortholog (such as co-purification with the complex, or at least interaction with a Mediator subunit in a GST-pulldown or yeast-two-hybrid experiment), I don’t think this is sufficient evidence for the claim made in the title "The nematode homologue of Mediator complex subunit 28, F28F8.5, is…”. The gene is certainly important for development, but may, or may not, be a MED28 protein. If the authors were to test this, MDT-28 is predicted to directly bind MDT-30, -29, and -27 according to Tsai et al Cell 2014, and these interactions should be detectable.

Additional comments

Minor comments and typos:
- General: F28F8.5 is a WormBase Gene ID and not a gene name and should NEVER be italicized.
- line 161: “as”
- line 175: “pCJ90"
- line 194: "and or"
- line 195 "Phusion High/Fidelity”
- line 198: “microinjecitons,”
- line 229: “levamizole” should be “levamisole”
- line 243: define “TTTR”
- line 282: “sequences”
- line 377: the statement "This suggests that Med28 may be able to bring cytoplasmic regulatory interactions towards the regulation of gene expression” is complete speculation and should be labeled as such. It is alternatively very well possible that the protein performs dual functions that do not overlap.

Reviewer 2 ·

Basic reporting

No Comments

Experimental design

1. The manuscript lacks controls for figure 4 and 5. While the authors discuss the generation of control siRNA, the results of such an injection are not discussed. It is important to have images of controls in figure 4 to be able to judge the severity of the phenotype. This is also important for readers unfamiliar with C.elegans biology. The authors should also consider re-arranging figure 4 to make it easier to follow.
2. Similarly in figure 5, the authors should include data from GFP::F28F8.5/ GFP (heterozygous) animals and compare it to data from the homozygous animals to be able to highlight the phenotype and the severity of the phenotype.
3. Since the authors state that the GFP fused F28F8.5 is faint, they should consider using an antibody to enhance the signal.

Validity of the findings

1.The authors state that F28F8.5 is localized in both the nucleus and the cytoplasm. However the magnification in figure 3 (B, C, F and G) is insufficient to make that claim. Further, the authors should co-label with DAPI or another nuclear dye to confirm the presence or absence of GFP signal from the nucleus. Figure 3I appears to have cells without nuclear expression of GFP. This should be validated by co-labeling with DAPI.
2.Similarly the authors state that GFP can be seen in the nuclei of the developing vulva in figure 2N. At this magnification, I find it hard to visualize expression in single cells, let alone specific cellular compartments.
Again in Figure 5, the images shown cannot be used to interpret the localization of GFP in these cells. I am unable to visualize nuclei or the cytoplasm. The authors should replace these images with confocal images at a higher magnification and preferably with worms stained with a GFP antibody and DAPI to mark the nuclei.
4.The authors discuss the phenotypic differences between the siRNA injected animals to those carrying a null mutation. The authors hypothesize that the maternal gene product in the null animal is sufficient to allow normal development. In addition, the authors should also discuss if the increased severity they observe in the case of the siRNA injections could be attributed to the targeting of the multi gene pre-mRNA that F28F8.5 is predicted to be a part of (https://www.ncbi.nlm.nih.gov/pubmed/10545456).

Additional comments

The authors should provide high quality images to prove that F28F8.5 is indeed expressed in both the nucleus and the cytoplasm as outlined above.
Controls are needed for both figure 4 and 5.
Minor points
1.Figure legend for Figure 5 has a typo (CRISP instead of CRISPR)

Reviewer 3 ·

Basic reporting

The authors found an important mistake in the annotation of MED28 in the worm. Previous published work from this group pointed out a missannotation for the Mediator subunit 28 in the worm genome now renamed perilipin. The authors then asked the question if C.elegans lacks a MED28 protein and go ahead and search for it.
The topic is important as it provides more information in the regulation of PolII transcription by Mediator in metazoans.

I think the paper should be published: experimental design it is very good, techniques used are up-to date and well controlled. Writing could be better organize and smoother.

But I think there is some major tuning-down of claims if other experiments can not be performed. So I suggest revisions.

At the end, the sequence analysis is the only suggestion that F28F8.5 works in the mediator complex. Too much emphasis is made on the dual nuclear/cytoplasmic localization as a proof of orthology. And actually that subcellular localization it is not very clear to me in some cases.
From what I understood subcellular localization of this mediator subunit in other organism hasn’t even been proven to be important (I might have missed that).

Co-IP with some of the other subunits of the head of the complex will be a proof that the protein forms part of the Mediator complex.
Rescue with fly or human protein?
It should be clearly stated in the discussion that these type of experiments will confirm F28F8.5 as MED28 (if any of these experiments can’t be done)

At least it should be correlated in the discussion instead of a lengthy discussion of the origin of cytoplasmic factors that can alter transcription (for instance b-catenin being one of the most known and with a profound function in development) with the phenotypes of Mediator function in the worm (reviewed in Grants et al NAR 2015 43 4:2442).

Looking forward for a revised version.
Please check "General comments to authors" section to answer my concerns. Thanks

Experimental design

It is good.

Validity of the findings

Findings are important to the field.
Some tunning-down of conclusions. Please check "General comments to authors" section

Additional comments

What needs to be included:

-Supplemental figure with locus indicating CRISPR tagging, primers for promoter fusion. Promoter fusion (just upstream sequences, introns? Not reader friendly as it is.)
Primers and products expected for hets and homo sterile mutants.

-PAM sequences selected.

-Table of strains generated in the paper following Wormbase nomenclature for edited strains



-Fig 2 Should be better organized and better explained in text for the expression pattern. A picture of each of the cell types mentioned in text must be shown with clear Nomarski for nuclei identification and fluorescence.

A panel of embryo development can be easily shown: before gastrulation, gastrulation, bean, coma, 2-fold and 3 fold. (higher magnification) H,J,K too exposed. It is not very clear to me that it can be claimed when referring to this picture that the expression is nuclear and cytoplasmic.
What about the pharynx, maybe it is easier there to check.
What about in neurons? Do you see fluorescence in the processes? Hypodermis? Seam cells?
Larval stages can be used to focus on selected cell types: gonad in L1, better close up to the head, VNC neurons, close-up of vulval muscles, Anchor cell.

Maybe it is only cytoplasmic and nuclear in few cell types. (Matus et al 2010 show clear nuclear expression only in the anchor cell).

Are all of these animals imaged homozygous for GFP::F28F8.5?(after heatshock)

-Fig 3. Embryo FILM it is not clearly shown. Higher mag is needed. Why rely so intestine cells that have so much granule autofluorescence. Need a control of a protein that it is also localized in nucleus and cytoplasm for positive control and cutoff of FILM numbers
Why not use epidermal cells?
No expression is seen in excretory cell with CRISPR tagging but yes with the promoter fusion although a phenotype is seen if Fig5. Suggest an explanation

-Phenotypes by CRISPR would be better before RNAi phenotypes description.


How this expression pattern compares with the promoter fusion? Introns? 3’utr?
Figure S1 cytoplasmic and nuclear localization is clearer where it can be also pointed out that is excluded from the nucleolus.
But always choosing gut cells I think it is not the best example with the granules autofluorescence. Can you show this in other cell type? Embryos are too faint to see

Is expression pattern in males the same?

Description of phenotypes should be clearer. Is it only sterility the phenotype observed in zygotic homozygous mutants? Is spermatheca formed in Hermaphrodites? Staining with DAPI may give a clearer idea.
Couldn’t find RNAi phenotype by feeding described in M&M.



Other comments:

In general, the writing could be smoother.

I think the paper will read better if the whole expression pattern is first (CRISPR and promoter fusion) and then bring up F28F8.5 as an essential gene with the RNAi experiments and CRISPR in one section being clearer in the phenotype description.


-Couldn’t find legend of Fig. S1

Lines 280-281: I think homologs and not paralogs should be used. Otherwise mention how many paralogs in each species.


Lines 287-288. If Insecta is excluded how do you get the Drosophila homolog after the query?


Line 360: what is the evidence that the affected gonad development is somatic and germline. Description of phenotypes should be more precise. Link between

Line 303: I would use suggests instead of indicates.

Line 361: how is it know that it is undifferentiated tissue.

---

## Round 0.2 · accepted · Accept

Please make a few minor copy-editing type corrections, as per reviewers 1 and 3, at the levels of lines 406, 492, 500 and the figure legends.

·

Basic reporting

Excellent.
Line 492 of the revised manuscript: "CDK18" should be changed into "CDK8"
Line 500: "the copy numbers of individual Mediator subunits identified by quantitative proteomics..."; I think this is usually referred to as "abundance", not "copy numbers", a term that refers to DNA

Experimental design

Excellent.

Validity of the findings

Excellent. The physical interaction data really convince me that this is a valid, and very interesting finding.

Additional comments

Congrats on a very nice paper.

Reviewer 2 ·

Basic reporting

Noi comment

Experimental design

No Comment

Validity of the findings

The revised experiments with the addition of the requested controls increase the soundness of the authors interpretations.

Additional comments

The authors have commendably answered all reviewer requests and concerns and in my opinion this manuscript is now suitable for publication.

Reviewer 3 ·

Basic reporting

Previous published work from this group pointed out a missannotation for the Mediator subunit 28 in the worm genome now renamed perilipin. The authors then asked the question if C.elegans lacks a MED28 protein and go ahead and search for it.
The topic is important as it provides more information in the regulation of PolII transcription by Mediator in metazoans.

Experimental design

Very good.

Validity of the findings

The authors did an amazing effort to address all concerns raised in the first submission. The manuscript has improved enormously. I think it should be published, although I will raise still some issues to be addressed.
(please see General comments for the authors)

Additional comments

Although the imaging has improved a lot I’m still not super convinced about the dual subcellular localization in all cells, throughout the life cycle. (why fig3 H and J the embryo there is not blue?)

I do believe in early embryos shown in adult worms (and in the bean embryos showed in fig 3c) there is cytoplasmic presence of MDT-28 but it is not something that happens in every cell type especially in the adult worm (except maybe pharynx). I’m not saying there is not a relevant function for this cytoplasmic localization in the embryos.
Just go with the wording as far as the images tell us. I’m not asking for more experiments .
The authors did already a good job to opening up many more questions to be addressed in the future.
For me it is clear that in 2-fold embryos with the CRISPR tagged homozygous allele there is not much, if any, cytoplasmic GFP, shown by FLIM.
That’s why in line 384 it is not quite true. I think it should be rephrased not to be so categorical that GFP can be found in both compartments throughout the life cycle of the worm. I think that finding places where it is in both compartments with this allele shows that the cytoplasmic localization could be relevant for MDT-28 function. Remember these are only 2 copies compared to a multi-copy array.

Line 406: Excretory canal cell (not channel) “excretory cell and its channels”

Line 406: To say it is not expressed in the excretory canal cell it is not quite right. It should say “we couldn’t detect expression”. Just as a comment, the level of GFP in the canals are very weak compared with the nuclear levels in the translational fusion, did you check in the CRISPR tagged homozygous allele directly the excretory canal cell nucleus? It is easy to find by Nomarski. Just saying that you may not see the canals because GFP levels are very low in the edited allele.

Fig6D arrow in front of arrowhead shows an oocyte to me not a gut cell.

Figure 2: for NPR panels I would suggest a close up of the vulva and the head. At the actual magnification shown it’s not clear. Supp fig 2 panel D should be moved here. Crop gut, orient head A->P, you may increase a bit brightness. It is hard to see cytoplasmic presence in neurons, glia and hypodermis in the head. I can see some in pharyngeal muscle cells maybe. Definitely not detected in the nerve ring.

Figure 4: sensory not sensoric
Group male panels, WT and worms with sensory ray defects

Figure 5 legend: spermatheca

Line 388 authors could add here that there is a clear maternal contribution so then it fits with the phenotypic analysis (just as a suggestion)

Lines 621/622: male rays have a lot of autofluorescence did you check in WT animals if this fluorescence is absent? Then the claim of cytoplasm function because the protein is still there, it is not valid. It has an effect indeed because of the phenotype but we don’t know where is acting.

Lines 625/626: I’m a bit confused here. Authors don’t see elevated expression in the cytoplasm of the anchor cell as Matus et al with the Nter CRISPR tagged allele. What about the translational fusion? It is Cter tagged (line61).
We know it has a phenotype in that cell so even if it is not because of higher levels in the cytoplasm also it could be interesting to speculate that MDT-28 could have some specific function regulating target genes involved in translocation of basal membranes. Mediator subunits had been linked to transcription dependent on specific TFs in different systems (e.g. Nature 442:700–704, PNAS 105:18-6644–6649). So maybe there is a TF interacting with MDT-28 having targets involved in this very important developmental process.

Supp Fig1 legend: “or is found accented in nuclei”, didn’t understand this

Supp fig3: reference or protocol for DAPI without fixation.